# LANGUAGE MODEL PRE-TRAINING FOR HIERARCHICAL DOCUMENT REPRESENTATIONS

## ABSTRACT

Hierarchical neural architectures can efficiently capture long-distance dependencies and have been used for many document-level tasks such as summarization, document segmentation, and sentiment analysis. However, effective usage of such a large context can be difficult to learn, especially in the case where there is limited labeled data available. Building on the recent success of language model pretraining methods for learning flat representations of text, we propose algorithms for pre-training hierarchical document representations from unlabeled data. Unlike prior work, which has focused on pre-training contextual token representations or context-independent sentence/paragraph representations, our hierarchical document representations include fixed-length sentence/paragraph representations which integrate contextual information from the entire documents. Experiments on document segmentation, document-level question answering, and extractive document summarization demonstrate the effectiveness of the proposed pre-training algorithms.

## 1 INTRODUCTION

While many natural language processing (NLP) tasks have been posed as isolated prediction problems with limited context (often a single sentence or paragraph), this does not reflect how humans understand natural language. When reading text, humans are sensitive to much more context, such as the rest of the document or even other relevant documents. In this paper, we focus on tasks that require document-level understanding. We build upon two existing separate lines of work that are useful for these tasks: (1) document-level models with hierarchical architectures, which include sentence representations contextualized with respect to entire documents (Ruder et al., 2016; Cheng & Lapata, 2016; Koshorek et al., 2018; Yang et al., 2016), and (2) contextual representation learning with language-model pretraining (Peters et al., 2018; Radford et al., 2018).

In this work, we show for the first time that these two ideas can be successfully combined. We find that pretraining hierarchical document representations raises several unique challenges compared to existing representation learning. First, it is not clear how to pretrain the higher level representations. Using existing techniques, it is possible to initialize the lower level of the hierarchical representation, to represent words in the context of individual sentences e.g. using ELMo (Peters et al., 2018), or, in addition, initialize sentence vectors that are not dependent on the context from the full document using Skip-Thought vectors (Kiros et al., 2015), but it is not possible to initialize the full hierarchical neural structure. Second, bidirectional representations are standard in state-of-the-art NLP tasks (Wu et al., 2016; Weissenborn et al., 2017), and may be particularly important for document-level representations due to the long-distance dependencies. However, current pretraining algorithms often use left-to-right and right-to-left language models to pretrain the representations. This restricts the expressiveness of the hierarchical representations, as left and right contextual information do not fuse together when forming higher-level hierarchical representations.

In this paper, we address these challenges by proposing two novel approaches to pre-train document-level hierarchical representations. Both methods pre-train representations from unlabeled documents containing thousands of tokens. Our first approach generalizes (Peters et al., 2018) to pre-train hierarchical left-to-right and right-to-left document representations. To allow the hierarchical representations to learn to fuse left and right contextual information from abundant unlabeled text, our

second approach is a new pretraining algorithm called *masked language model*, which allows efficient training of bidirectional hierarchical document representations.

We evaluate the impact of the novel aspects of our pre-trained representations on three tasks: document segmentation, answer passage retrieval for document-level question answering, and extractive text summarization. We first pretrain document-level representations on unlabeled documents and then fine-tune with task-specific labeled data and a light-weight task network. Experiments show that pre-training hierarchical document representations brings significant benefits on all tasks, and that pretraining the higher level of the document representation results in larger improvements than pre-training only the locally contextualized lower word level in most tasks. On the CNN/Daily Mail summarization task, we obtain strong results without using reinforcement learning methods employed by the previous state-of-the-art models. On the TriviaQA answer passage retrieval task, our model improves over a strong baseline (Clark & Gardner, 2018) by 6% absolute.

## 2 BACKGROUND

In this section, we review pretraining of contextual token representations using language models and discuss how the language model probability decomposition imposes constraints on the architectures.

**Language Model Pretraining**  Given a sequence of tokens $(x_1, \ldots, x_n)$, a representation module $V_\theta$ encodes each word into a vector: $V_\theta(x_1, \ldots, x_n) = (v_1, \ldots, v_n)$, where each word-level contextual representation $v_i$ could potentially depend on the whole sequence. Common choices for the representation modules are LSTMs, CNNs, and self-attention architectures (Hochreiter & Schmidhuber, 1997; Vaswani et al., 2017; LeCun & Bengio, 1998). A representation module is parameterized by an underlying neural architecture with parameters $\theta$.

The typical left-to-right language model aims to maximize the probability of observed sequences:

$$\prod_{t=1}^{n} P(x_t | x_1, \ldots x_{t-1}; \theta), \tag{1}$$

The pretrained parameters $\theta$ can then be used to initialize the model for a downstream task.

**Uni-directionality constraint**  The formulation of the language model implicitly imposes a directionality constraint on the $V_\theta$, which we term *uni-directionality constraint*. In Equation 1, the word $x_t$ can only depend on the (representations of) the previous words. Therefore, the contextual representations $v_1, \ldots, v_{t-1}$ can not depend on the representations of $x_t, \ldots, x_n$, resulting in the uni-directionality constraint on the representations.

One common type of representation modules that satisfy the uni-directionality constraint are unidirectional recurrent neural networks (RNNs). Given a sequence of words, a left-to-right LSTM model generates a sequence of representations $V_\theta(x_1, \ldots, x_n) = (\overrightarrow{h}_1, \ldots, \overrightarrow{h}_n)$ that satisfies the uni-directionality constraints, where $\overrightarrow{h}_t$ only depends on $\overrightarrow{h}_1, \overrightarrow{h}_2, \ldots \overrightarrow{h}_{t-1}$. Modules that satisfy the uni-directionality constraint can also be derived from self-attention, convolution, and feed-forward architectures (e.g. (Radford et al., 2018)).

As mentioned in the introduction, for many downstream tasks, bidirectional representations bring large improvements over uni-directional ones. Therefore, Peters et al. (2018), pre-trained two encoders using two language models: a left-to-right and a right-to-left model, using a left-to-right and a right-to-left LSTM encoder, respectively. The token representations are formed by the concatenation of outputs from the two uni-directional encoders. We term this style of language model pre-training L+R-LM. More specifically, the representation of the $t$-th word becomes $v_t = \left[\overrightarrow{h}_t \quad \overleftarrow{h}_t\right]$, where $\overleftarrow{h}_t$ is generated from a right-to-left LSTM. Tasks using these input representations must learn to merge the left and right contextual information from scratch using only the labeled data.

The general strategy we adopt in this paper is to pretrain the hierarchical representations on unlabeled data and then to fine-tune the representations with the task-specific labeled data. Therefore, the uni-directionality constraint will also impacts the task-specific networks.

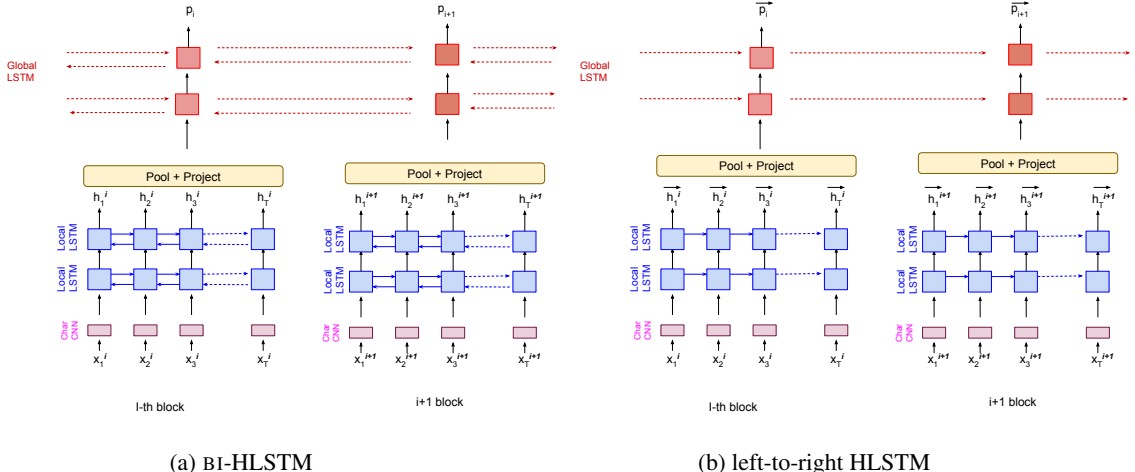

(a) BI-HLSTM          (b) left-to-right HLSTM

Figure 1: Document-level representations used in this paper. **(a)** Bi-HLSTM: the bidirectional hierarchical representation, which does not satisfy the uni-directionality constraint. **(b)** left-to-right HLSTM (L-TO-R-HLSTM): a uni-directional version of BI-HLSTM.

# 3 HIERARCHICAL DOCUMENT-LEVEL REPRESENTATIONS

As our goal is to learn fixed-length contextualized representations of text segments to be used in document-level tasks, we choose hierarchical neural architectures that build representations of tokens and text segments in context. Here we describe the specific architectures studied in this paper. The novel contribution of our work – pre-training such representations from unlabeled text, will be detailed in Section 4.

We choose LSTM-based architectures with a standard hierarchical structure that has been useful for capturing long-term context in document-level tasks (Serban et al., 2016; Koshorek et al., 2018). We experiment with two related document-level representations. The first one, called BI-HLSTM, is a standard hierarchical bidirectional LSTM encoder. BI-HLSTM does not satisfy the uni-directionality constraint and cannot be pre-trained using a left-to-right language model likelihood objective function. Our second document-level representation is L+R-HLSTM, which consists of two concatenated uni-directional versions of BI-HLSTM, and can be seen as a hierarchical document-level extension of ELMo (Peters et al., 2018). Both document-level representations view documents as sequences of text blocks, each consisting of a sequence of tokens. The text blocks can be sentences, paragraphs, or sections, depending on the task. Here we experiment with sentences and paragraphs as text blocks.

The BI-HLSTM architecture fuses left and right contextual information more effectively and directly corresponds to state-of-the-art architectures used for multiple NLP tasks.

BI-**HLSTM**    BI-HLSTM has a two-level hierarchy. Every block is first independently processed by a bidirectional LSTM (Bi-LSTM) to encode the local context. The contextual representations of the words in each block are then pooled and projected to a single vector to represent the block. The vector representations of the sequence of blocks in the document are then processed by a global Bi-LSTM, resulting in block representations contextualized by the entire document. The overall design is shown in Figure 1a, and the encoding procedure is formalized next.

A document $D$ consists of $K$ blocks $D = \{U_1, U_2, \ldots U_K\}$, and each block consists of a sequence of tokens where $U_i = \{x_1^i, x_2^i, \ldots x_T^i\}$.[1] For a block $U_i$, we start with character representations of the words, where $\bar{x}_t^i = \text{CNN}_{\text{char}}(x_t^i)$. Then character-based word representations are processed by the local Bi-LSTM, where $\{h_1^i, \ldots, h_T^i\} = \text{BI-LSTM}_{\text{Local}}(\{\bar{x}_1^i, \ldots, \bar{x}_T^i\})$.

---

[1] In our experiments, the value of $T$ depends on $i$. For simplicity, we slightly abuse notation and use same $T$ for all blocks.

Before generating the globally-contextualized representations, the encoder generates representations of all blocks by pooling the locally-contextualized word representations generated from the local encoder. The block representations are obtained using both max and average-pooling on the word representations, followed by a feed-forward transformation:

$$c_i = \text{FFNN}(\text{POOL}_{\text{max}}(\{h_1^i, \ldots, h_T^i\}); \text{POOL}_{\text{avg}}(\{h_1^i, \ldots, h_T^i\})).$$

The resulting local block representations $c_i$ are then processed by the global Bi-LSTM to generate document-contextualized block representations $p_i$: $\{p_1, \ldots, p_K\} = \text{BI-LSTM}_{\text{Global}}(\{c_1, \ldots, c_K\})$.

In downstream tasks, this document representation is used by inputting the text block contextual vector representations $\{p_i\}$ to light-weight task-specific networks.

L+R-**HLSTM** As previously mentioned, when using left-to-right language models to pre-train representations, the representation of the $i$-th word can only depend on the representations of the previous words $(i-1)$ words. Similarly, the representation of the $j$-th text block can only depend on the representations of the previous text blocks. To design a document-level representation similar to the one defined by the BI-HLSTM but that can still be trained using a uni-directional language model decomposition, we replace all of the bi-directional LSTMs in BI-HLSTM with either left-to-right or right-to-left LSTMs. The left-to-right version of the resulting hierarchical encoder is presented in Figure 1b.

Given a document $D = \{U_1, U_2, \ldots U_K\}$, let the encoding results of the left-to-right HLSTM be denoted as $\{\overrightarrow{p}_1, \ldots, \overrightarrow{p}_K\} = \text{L-TO-R-HLSTM}(D; \theta_l)$. We use another right-to-left hierarchical encoder to produce representations: $\{\overleftarrow{p}_1, \ldots, \overleftarrow{p}_K\} = \text{R-TO-L-HLSTM}(D; \theta_r)$. We then concatenate the final representations of the two encoders: $\hat{p}_i = \left[ \overrightarrow{p}_i \quad \overleftarrow{p}_i \right]$.

We term the resulting document-level encoder L+R-HLSTM. Note that the L+R-HLSTM combines the left and right information only at the top level, and there is no interaction between the two uni-directional contextualizers. While at the bottom-most layer, the BI-HLSTM similarly concatenates left and right information, the second local BI-LSTM layer can fuse the two directions. Similarly, the left and right contexts at the text segment level are fused at the second global BI-LSTM layer.

In principle, both BI-HLSTM and L+R-HLSTM can capture long distance dependencies via their hierarchical structure, but it is unclear whether such dependencies can be effectively learned from limited amounts of supervised data labeled for down-stream tasks of interest. In the next section, we discuss how such document encoders can be trained to provide rich text segment and token representations using large unlabeled document collections.

# 4 PRE-TRAINING HIERARCHICAL DOCUMENT-LEVEL REPRESENTATIONS

Here we propose language model-based methods for pre-training hierarchical document-level representations. In order to train the hierarchical representations with language models, our main idea is conceptually simple: we combine sentence representations contextualized with respect to the whole document *and* the token representations with respect to the sentence to perform missing word predictions. Intuitively, from the loss of word prediction, both the contextual sentence and token representations will be updated.

Note that there are interactions between the contextual sentence and token representations, as part of the contextual sentence representations are constructed from token representations. Therefore, the language model pretraining algorithms needed to be designed carefully. In Section 4.1, we present ways to pre-train uni-directional hierchical document-level representations. In Section 4.2, we propose a language prediction model that can pre-train bi-directional representations.

## 4.1 UNI-DIRECTIONAL REPRESENTATIONS

Standard language model pre-training (L+R-LM) can be used to pre-train the uni-directional token-level representations in local text block context. Starting from Equation 1, and using a standard LSTM language model, we can define the predictive probability of the next word given prior text block-internal context as: $P_{\text{L-LM}}^{\text{local}}(x_t^i | x_{t-1}^i, \ldots x_1^i) = P(x_t^i | \overrightarrow{h}_{t-1}^i)$, and train the representations us-

ing this language model. As in Peters et al. (2018), this only pre-trains token representations in local context and does not provide a way to derive contextualized single vector text block representations.

Here we propose to use hierarchical language models to pre-train uni-directional hierarchical document-level representations. We refer to this algorithm as L+R-LM$^{\text{global}}$. When pre-training with hierarchical language models, we treat the uni-directional document-level representation in Figure 1b as the representation used by a hierarchical language model and use entire unlabeled documents to pre-train it.

The hierarchical language model predicts the next word in a document using the contextual encoder representations. The left-to-right hierarchical model can use the contextual information from previous tokens in the same document to predict the next word. The previous tokens with respect to a current token can be categorized into two sets: the preceding tokens in the same text block, and all tokens in the *preceding* text blocks. More specifically, the predictive probability for the next token is defined as: $P_{\text{L-LM}}^{\text{global}}(x_t^i | x_{t-1}^i, \ldots x_1^i, U_{i-1}, \ldots, U_1) = P(x_t^i | \overrightarrow{h}_{t-1}^i, \overrightarrow{p}_{i-1})$.

Therefore, the objective function for L+R-LM$^{\text{global}}$ is as follows:

$$\min_{\theta_l, \theta_r} \sum_{t,i} \log P(x_t^i | \overrightarrow{h}_{t-1}^i, \overrightarrow{p}_{i-1}, \theta_l) + \log P(x_t^i | \overleftarrow{h}_{t+1}^i, \overleftarrow{p}_{i+1}, \theta_r),$$

where we use a feed-forward network to combine the information from $(\overrightarrow{h}_{t-1}^i, \overrightarrow{p}_{i-1})$ and $\overleftarrow{h}_{t+1}^i, \overleftarrow{p}_{i+1}$, respectively. These feed-forward networks are not used in down-stream tasks.

## 4.2 BI-DIRECTIONAL REPRESENTATIONS

We have shown a method to pre-train uni-directional encoders with hierarchical language models. However, it is still not clear how we can pre-train representations that do not satisfy the uni-directionality constraint from unlabeled text. Here we present text prediction models we term masked language models (MASK-LM), which alleviate the architectural constraints imposed by standard language model pre-training. They can be seen as a special form of denoising auto-encoders (Vincent et al., 2008), where the decoder network has minimal parameters, the only type of noise is word masking, and the decoder only makes predictions for positions with noise (from the terminology in that paper, full emphasis is on reconstructing the corrupted dimensions).

Figure 2 demonstrates the method of training representations with MASK-LM. For a given document, we first generate a set of random indices and mask out (hide) the words at the corresponding positions in the document. After a document has been masked, we provide it as input to a document-level encoder to produce contextual word and text block representations. These contextual representations are used to predict the words at the masked positions.

More formally, let $Z$ denote a set of indices for the masked words. Let $\zeta(D; Z) = \{\zeta(x_1^1), \zeta(x_2^1), \ldots, \zeta(x_t^i), \ldots \zeta(x_T^K)\}$ indicate the masked word sequence, where:

$$\zeta(x_t^i) = \begin{cases} \square, \forall (i,t) \in Z, \\ x_t^i, \text{Otherwise.} \end{cases}$$

Figure 2: The general procedure of using masked language model to train any encoders. Note that masks are randomly generated. See text for more details.

Note that $\square$ is a special symbol to indicate that the word is masked out (hidden). Given that $Z$ is generated randomly, masked language models essentially optimize the following objective function where $\theta$ represent the parameters of the representation:[2] $\max_\theta \left( \mathbb{E}_Z \left[ \sum_{(i,t) \in Z} \log P(x_t^i | \zeta(D; Z), \theta) \right] \right)$.

---

[2]Note that the model does not learn a probability distribution over possible texts and is thus not formally a language model. Nevertheless, for ease of exposition and in line with prior work on learning representations from unlabeled text (Collobert & Weston, 2008) we refer to this model as a language model.

Unlike the auto-encoder method of Hill et al. (2016), in MASK-LM we assume the probabilities of all masked words in a sequence can be predicted independently and do not employ a separate auto-regressive decoder network.

There are two ways to apply MASK-LM to pre-train parts of the BI-HLSTM. Analogously to the local setting of L+R-LM, we could pre-train with MASK-LM using local context. We define the probabilities of masked words to depend only on the locally contextualized token representations, and apply MASK-LM to only pre-train the local bi-LSTM by using: $P_{\text{MASK-LM}}^{\text{local}}(x_t^i|\zeta(D;Z);\theta) = P(x_t^i|h_t^i;\theta)$, where the $h_t^i$ is the word-level contextual representation for masked word $x_t^i$. Note that if $(i,t) \in Z$, $\zeta(x_t^i) = \square$. Therefore, $h_t^i$, which is generated by the bi-directional local LSTM encoder, needs to carry the contextual information in order to recover the masked word $x_t^i$.

Similarly, in the MASK-LM$^{\text{global}}$ setting, the predictive probabilities are defined as: $P_{\text{MASK-LM}}^{\text{global}}(x_t^i|\zeta(D;Z);\theta) = P(x_t^i|h_t^i,p_i;\theta)$, where we use another feed-forward network to combine the information from $h_t^i$ and $p_i$.

## 5  DOWN-STREAM TASKS

We perform experiments on three document-level downstream tasks, which require predictions at the level of text blocks in full documents. The predictions are made using light-weight task specific networks which take as input single-vector document-contextualized text block representations $p_i$, generated by the hierarchical document encoders described in Section 3. We briefly define the downstream tasks and the task-specific architectures used. We provide additional details in the Experiments section.

**Document Segmentation** In the document segmentation task, we take as input a document represented as a sequence of sentences. The goal is to predict, for each sentence, whether it is the last sentence of a group of sentences on the same topic, thereby segmenting the input document in topical segments. We use a feed-forward network with one hidden layer with RELU activations on top of sentence representations $p_i$ to define scores of the task specific labels (segment boundary or not).

**Answer Passage Retrieval** Given a document and a question, the task of answer passage retrieval is to select passages from the document that answer the question. While most QA systems focus on predicting short answer spans given a relevant answer passage, answer passage retrieval is a prerequisite task for open-domain question answering. While there are a multitude of architectures possible for this task, we propose a modular approach that enables extremely fast answer passage retrieval during serving time. Our model works as follows. The given document is encoded and its contextual passage representations $p_i$ are generated. The questions are treated as documents consisting of a single sentence and are similarly encoded in representations $q$. Inspired by Peters et al. (2018), instead of using the representations $p_i$ directly, we use a linear combination of the layers of $p_i$ with positive weights summing to 1: $p_i' = \sum_l s_l p_{i,l}$, where $s_l$ is a task-specific learned scalar weight for the $l$-th layer, and $p_{i,l}$ represents the $l$-th layer of the representations of the $i$-th block. To score a passage-question pair, we use the following architecture using a learned task-specific network combining the following features: $s_0(p_i, q) = \langle p_i', q' \rangle$ is a dot product of the passage and question representations, $s_1, \ldots, s_5$ are five light-weight features such as paragraph position and tf-idf borrowed from Clark & Gardner (2018). These features are sent to a feed-forward network with one hidden layer with RELU activations to generate the final score.

**Extractive Summarization** Given a document, the task of extractive summarization is to select representative sentences to form a summary for the whole document (Kupiec et al., 1995). Our summarization model is quite similar to our answer passage selection model with two key differences. First, we add another randomly initialized LSTM on top of all sentence representations to form refined sentence representations. We found that the extra layer generally improves results. Second, we perform max-pooling over the contextual representations of all sentences $p_i'$ to form a fixed-length vector representation $d$ for the entire document. The representation $d$ is then treated like the "question" representation in the passage selection task. Third, instead of using a dot-product as the scoring function, for each sentence, we concatenate $d$ and $p_i'$ to form a combined vector which is sent to a feed-forward network with one hidden layer with RELU activations to generate the final score. As for the Answer Passage Retrieval task, we add several task-specific features such as the position of the sentence in the document and the number of words in the sentence.

## 6 EXPERIMENTS

The main goal of our experiments is to examine the value of unsupervised pre-training of hierarchical document-level representations, focusing on the novel aspects of our methods. We first present the experimental settings and pre-training details in Section 6.1. We then present and analyze the experimental results on the three tasks.

Among the pre-training methods, L+R-LM$^{\text{global}}$, MASK-LM and MASK-LM$^{\text{global}}$ are proposed in this paper. The local version of L+R-LM is similar to ELMo (Peters et al., 2017), but its effectiveness when used to initialize document-level hierarchical representations has not been studied before.

### 6.1 SETTINGS

We use standard LSTM cells for our sequence encoders. For word embeddings, we use character-based convolutions similarly to (Peters et al., 2018), and project the embeddings to 512 dimensions. Our local encoder is a 2-layer LSTM with 1024 hidden units in each direction. Our global encoder is another 2-layer LSTM with 1024 hidden units.

Since L+R-LM requires the representation to satisfy the uni-directionality constraint, while there are no directionality constraints on the representations when pre-training with MASK-LM, in the experiments we always pair each encoder with its natural pre-training method: we pre-train BI-HLSTM with MASK-LM, and L+R-LM with L+R-HLSTM. For each of the two encoders, we compare no pre-training on unlabeled data, versus pre-training at the local sentence level only, versus pre-training at the global document level.

If we initialize the BI-HLSTM representation with locally pre-trained MASK-LM, this means that we initialize the local Bi-LSTM and learn the global Bi-LSTM only based on the labeled down-stream task data. If we initialize both the local and global Bi-LSTMs of BI-HLSTM with MASK-LM$^{\text{global}}$ pre-trained models, only the task-specific feed-forward network is randomly initialized, and all parameters are fine-tuned with the labeled set. When comparing hierarchical (global) and non-hierarchical (local) pretraining methods, we use the *same* document-level representation architecture for the downstream tasks, pre-train on the same amount of unlabeled text, and then fine-tune with the same task-specific network architectures.

In all tasks, we also included comparisons to skip-thought vectors (Kiros et al., 2015), where the model is learned to construct sentence embeddings.[3] Additionally, we compare to EMLO$_{\text{pool}}$, which forms a representation for a text block by pooling the token-level representations generated by the model pretrained in (Peters et al., 2018). Perone et al. (2018) have shown that EMLO$_{\text{pool}}$ produces strong sentence representations and outperforms many other sentence representation models including skip-thought vectors.[4] At a high level, the contextualized token representations obtained by our local L+R-LM implementation are similar to the ones obtained by ELMo. However, ELMo was trained on a larger dataset and uses higher-dimensional hidden vector representations.

To pre-train the hierarchical representations, we use documents in Wikipedia and filter out ones with fewer than three sections or less than 800 tokens. We sample from the filtered document set and form a collection containing approximately 320k documents with about 0.9 billion tokens. We select the most frequent 250k words as the vocabulary for all models. We limit documents to at most 75 sentences and limit each sentence length to 75 tokens. Sentences are used as text blocks for pre-training and pre-trained models are applied to both sentence-block and paragraph-block documents in the down-stream tasks. To capture the full hierarchical information, we did not use truncated back-propagation-through-time (tBPTT) (Rumelhart et al., 1985), but used full back propagation on documents with up to 5k tokens. For MASK-LM, we randomly mask 20% of the words in the documents for training. All of the down-stream task experiments are performed using the same set of pre-trained models.

### 6.2 DOCUMENT SEGMENTATION

We create labeled data for the segmentation task by taking Wikipedia articles and forming the labels using the section information. Given a sentence in a Wikipedia article, the label of the sentence

---

[3]We use the bi-directional model for the skip-thought vectors.

[4]We use the "all layers" and "original" settings according to (Perone et al., 2018).

| Pre-training | MASK-LM BI-HLSTM | L+R-LM L+R-HLSTM |
|---|---|---|
| No | 42.0 (0.0) | 41.7 (0.0) |
| Local | 50.6 (8.6) | 48.4 (6.7) |
| Global | **51.8** (9.8) | **54.9** (13.2) |
| LSTM+ELMo$_{pool}$ | 44.6 | |
| LSTM+Skip-Thought | 46.0 | |

(a) Document Segmentation

| Pre-training | MASK-LM BI-HLSTM | L+R-LM L+R-HLSTM |
|---|---|---|
| No | 77.24 (0.0) | 77.20 (0.0) |
| Local | 79.17 (1.9) | 78.36 (1.2) |
| Global | **79.92** (2.7) | **79.57** (2.4) |
| (Clark & Gardner, 2018) | 73.31 | |

(b) Answer Passage Retrieval

Table 1: Downstream task performance for the segmentation and answer passage retrieval tasks. The improvements over corresponding baselines are indicated in brackets. **(a)** F1 on document segmentation. **(b)** Precision at one (P@1) for answer passage retrieval in TriviaQA-Wiki. We compare our models to the answer passage retrieval module developed by Clark & Gardner (2018).

is positive if it is the last sentence of a Wikipedia section. To ensure fair comparisons, the section boundary information is never used during pre-training. We select 5k documents each for training, development, and test sets. In our dataset, only 5% of the sentences are positively labeled; we use F1 score as the evaluation metric due to this imbalanced class distribution. All reported experimental results for the segmentation task are averages of five different training runs on the task-specific labeled data.

**Results** The experimental results for the segmentation experiments are in Table 1a. Our baseline system that does not use any pretraining of representations is similar to the one proposed in Koshorek et al. (2018) with the difference that our system uses character information to generate the word embeddings. Note that pretaining results in substantial improvements over the baseline systems in all settings. We next compare pre-training with hierarchical language models (the global setting) versus pre-training with non-hierarchical language models (the local setting). Global pre-training of the hierarchical document-level representations improves upon local pre-training of the sentence-contextualized token representations. In the case of the L+R-HLSTM encoder paired with L+R-LM$^{global}$, the improvement is over 6% F1. For this task, we found that L+R-LM$^{global}$ is much better than MASK-LM$^{global}$. We hypothesize that by predicting words in the next sentence, the L+R-LM$^{global}$ model is more sensitive to the topical changes between sentences. The Table also shows results of experiments with a setting where we applied another Bi-LSTM on top of the EMLO$_{pool}$ and skip-thought vectors; as seen both methods perform worse than our approaches.

## 6.3 ANSWER PASSAGE RETRIEVAL

We apply the document-level encoders to the task of answer passage retrieval in the next set of experiments. To evaluate the impact of pre-training a document-level encoder for this task, we use the Wikipedia subset of the TriviaQA dataset (Joshi et al., 2017), a large scale distantly supervised question answering dataset. While TriviaQA was originally designed for short answer detection, we use it to evaluate the answer-passage selection task, as it takes the full documents as input and requires retrieving passages. [5]

We use the paragraph breaker implementation used in Clark & Gardner (2018). As in other work on TriviaQA, we set 400 to be the maximum number of tokens in a paragraph during training. We also use 400 tokens in testing. We report the top-1 passage selection accuracy on the development set as our main metric. As the TriviaQA test data labels are not publicly available, we cannot obtain passage accuracy test metrics and thus do not report these.

The passage retrieval results are in Table 1b. Note that pre-training leads to substantial improvements in all settings. For both MASK-LM and L+R-LM, global pre-training of the full hierarchical representations improves upon local pre-training of the lower-level contextual token representations. Our models improve upon the passage selection model proposed by (Clark & Gardner, 2018) by more than 6% absolute. The latter work focuses on evaluation of their end-to-end short answer selection

---

[5]We did not study the TriviaQA Web subset as the first paragraph baseline has a strong accuracy of 83% for it, reducing the importance of strong passage selection systems in this dataset.

| Pre-training | MASK-LM BI-HLSTM | L+R-LM L+R-HLSTM |
|---|---|---|
| No | 36.9 (50.5) | 37.3 (49.8) |
| Local | 37.0 (50.6) | 37.2 (50.0) |
| Global | **37.4 (51.5)** | 37.4 (50.7) |

(a) Evaluating the impact of different pretraining algorithms. Rouge-L only is shown for brevity here. The number in the bracket is the accuracy of the top-rank sentence.

| Pre-trained Vectors | P. Retrieval (P@1) |
|---|---|
| MASK-LM$^{global}$ | **66.2** |
| L+R-LM$^{global}$ | 63.3 |
| L+R-LM$^{local}_{pool}$ | 53.8 |
| EMLO$_{pool}$ | 56.8 |
| Skip-Thought | 37.2 |

(b) To inspect the quality of the pretrained representations, we show a zero-shot setting for the answer passage retrieval task.

Table 2: Analysis of the impact of different pretraining methods on document summarization and zero-shot answer passage retrieval.

| Models | R-1 | R-2 | R-L |
|---|---|---|---|
| LEAD (See et al., 2017) | 39.6 | 17.7 | 36.2 |
| NeuralSum (Cheng & Lapata, 2016) | 35.5 | 14.7 | 32.2 |
| SummaRuNNer (Nallapati et al., 2017) | 39.6 | 16.2 | 35.3 |
| REFRESH (Narayan et al., 2018) | 40.0 | 18.2 | 36.6 |
| Bottom-up Summarization (Gehrmann et al., 2018) | 40.7 | 18.0 | 37.0 |
| MASK-LM$^{global}$ (Ours) | **41.0** | **18.6** | **37.4** |
| LSTM+Elmo$_{pool}$ | 40.6 | 18.4 | 37.0 |
| LSTM+Skip-Thought | 40.5 | 18.2 | 36.9 |
| NeuSum (Zhou et al., 2018) | 41.6 | 19.0 | 38.0 |

Table 3: Comparative evaluation of our summarization models with respect to other extractive summarization systems. ROUGE-1 (R-1), ROUGE-2 (R-2), and ROUGE-L (R-L) F1 scores are reported. The first part of the table indicates the system results with non-autoregressive sentence selection models. The second part of the table shows the performance with fresh trained LSTM with pretrained embeddings. NeuSum (Zhou et al., 2018) is included in the third part as it is specifically trained to score combinations of sentences with auto-regressive selection models.

system and does not report passage retrieval performance for TriviaQA-Wiki. We measured the performance using the authors' codebase[6].

**Analysis** Our answer passage retrieval model uses a dot-product between the question and passage representations to score passages. If we restrict our models to not use additional features apart from this dot product of the question and passage representations, we can derive task-specific models in a "zero-shot" setting, where we do not use labeled data from TriviaQA to fine-tune pre-trained representations. We use this zero-shot setting to further investigate whether the pre-trained representations have learned to assess the semantic similarity between questions and passages based on unlabeled text. The results are presented in Table 2b. In this setting, MASK-LM$^{global}$ achieves 66.2, outperforming L+R-LM$^{global}$, EMLO$_{pool}$ and skip-thought vectors. The max-pooled version of our local L+R-LM method under-performs EMLO$_{pool}$, which is explained by the difference in size of the pre-trained representations.

## 6.4 EXTRACTIVE DOCUMENT SUMMARIZATION

We use the CNN/DailyMail dataset (Hermann et al., 2015) to evaluate our summarization model. We follow the standard data pre-processing and evaluation settings for extractive summarization from (See et al., 2017), and use ROUGE (Lin, 2004) as the evaluation metric.

The results of our summarization system are presented in Table 3. Compared to other extractive summarization systems, our model improves upon the previous model (Narayan et al., 2018) by 0.9

---

[6]https://github.com/allenai/document-qa

in ROUGE-1 and 0.7 in ROUGE-L, without using the sophisticated reinforcement learning techniques employed by that work. Our model does not perform as well as NeuSum (Zhou et al., 2018), which employed an auto-regressive sentence selection model. Recently, an abstractive summarization system was able to outperform extractive ones (Paulus et al., 2017). We expect that the novel aspects of our work can also bring improvements in that non-extractive setting and are complementary to the use of auto-regressive sentence selectors. We also compared our models to models using $\text{EMLO}_{\text{pool}}$ and skip-thought vectors with Bi-LSTM. As seen these methods are also competitive, but are not as strong as our global hierarchical pre-training approach.

The impact of pretraining is analyzed in Table 2a. For brevity, we only include ROUGE-L scores, but ROUGE-1 and ROUGE-2 show similar trends. We also show the accuracy of the top-rank sentence. As for answer passage retrieval, $\text{MASK-LM}^{\text{global}}$ is the best overall pretraining algorithm.

# 7 RELATED WORK

Prior work in *supervised* learning of neural representations for full documents has shown the effectiveness of hierarchical neural structures, which contain representations of both the sentences and their component words in a two-level hierarchy. In the lower hierarchy level, the contextual word representation is formed with respect to the sentence. In the higher, the contextual sentence representation is formed with respect to the document. Such structures enable the models to integrate long-distance context from the documents and have been used for labeling sentence sentiment (Ruder et al., 2016), document summarization (Cheng & Lapata, 2016), text segmentation (Koshorek et al., 2018), and text classification (Yang et al., 2016), *inter alia*. These hierarchical neural representations have been largely learned based on task-specific labeled data, posing a challenge for applications with a limited number of annotated examples.

Unsupervised pretraining for hierarchical document representations has received relatively little attention despite the recent success on language model pre-training (Peters et al., 2018; Radford et al., 2018). Prior work has shown how to use unlabeled text to pre-train representations of individual words, e.g. (Mikolov et al., 2013), or flat (relatively short) sequences of words, where each word representation is contextualized with respect to the sequence (Peters et al., 2017; 2018; Salant & Berant, 2018; Radford et al., 2018). Similarly, fixed-length vector representations of full sentences/paragraphs have also been pre-trained from unlabeled text (Le & Mikolov, 2014; Kiros et al., 2015; Dai & Le, 2015; Logeswaran & Lee, 2018), where the sentence/paragraph representations are generated based on the content of the sentence/paragraph alone, ignoring other sentences in the document.[7]

Li et al. (2015) showed how to train hierarchical document representations from unlabeled text using document auto-encoders but did not use such representations in extrinsic downstream document-level tasks. To the best of our knowledge, no prior work has transferred hierarchical unsupervised document representations to downstream tasks, and evaluated the impact of pretraining local and global document-level contextualizers. In addition, bidirectional hierarchical representations have not been pretrained without a sophisticated decoder in an auto-encoder framework before.

# 8 CONCLUSION

In this paper, we proposed methods for pre-training hierarchical document representations, including contextual token and sentence/paragraph representations, integrating context from full documents. We demonstrated the impact of pre-training such representations on three document-level downstream tasks: text segmentation, passage retrieval for document-level question answering, and extractive summarization.

---

[7]Note that most approaches (e.g. (Kiros et al., 2015)), pretrain the representations by asking a model to predict the words in other sentences during training, but the encoders look at sentences in isolation to generate representations at inference time.

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
