# OpenReview forum: "Language Model Pre-training for Hierarchical Document Representations"
_ICLR.cc/2019/Conference_

### Official Review · AnonReviewer2 · 2018-11-02
**Good contribution with a few missing baselines and implementation details**

**Rating:** 6
**Confidence:** 4

**Review:**

In this work, the authors explore different ways to pre-train contextualized word and sentence representations for use in other tasks. They propose two main methods: a straight-forward extension of the ElMO model for hierarchical uni-directional language models, and a de-noising auto-encoder type method which allows to train bi-directional representations. The learned contextual representations are evaluated on three downstream tasks, demonstrating the superiority of the bi-directional training setting, and beating strong baselines on extractive summarization.

The method is clearly presented and easy to follow, and the experiments do seem to support the author's claims, but their exposition misses several important details (or could be presented more clearly). For the document segmentation task, are the articles taken from a held-out set, or are they contained in the pre-training set? For passage retrieval, is the representation the same or are the representations re-trained from scratch using paragraph blocks? What exactly are the other features (those can go in the appendix)? And for the extractive summarization task, how many sentences are selected? Is pre-training also done on Wikipedia, or are those representations trained on news text?

A comparison to non-contextualized sentence representations would also be welcome (SkipThought, InferSent, ElMO-pool for settings other than passage retrieval). Note also that the local pre-training is not equivalent to ElMO, as the later sees context form the whole document rather than just the current sentence.

It is interesting to see that contextualized sentence representations can be used and that the Mask-LM objective yields better results than L+R-LM, but these points would be better made if the above questions were answered.

---

> ### Author Response · Authors · 2018-11-14
> **Initial Response**
>
> We thank Reviewer 2 for the valuable comments. We will address the clarification suggestions in detail in an appendix in the paper and also provide brief explanations below.
>
>         * In the document segmentation task experiments, the dataset is sampled from the same set of articles used for pretraining. However, the labels for the segmentation task (section boundaries) are never used during pretraining.
>
>         * We pre-trained our representations using a sentence-based corpus, containing articles represented as lists of sentences. For passage retrieval, we directly apply the same model on the passages without re-training the models. The whole model is then fine-tuned using the passage retrieval labeled data.
>
>         * Other features for passage retrieval: these features were computed using the software \url{https://github.com/allenai/document-qa} of Clark & Gardner and are defined as follows: cosine similarity of tf-idf representations of passage and question, logarithm of scaled position of first word in the paragraph, indicator function of whether this is the first paragraph in the document, number of non-stop words in the paragraph that appear in the question, number of paragraph words that appear in the question but with a different case or that are stop-words.
>
>         * Currently we always select 3 sentences in the extractive summarization task. All of our tasks use the same pretrained model trained on Wikipedia.
>
>         * (Compare local setting with ELMo) In fact, the local setting is quite similar to ELMo,  as the existing ELMo models have been trained on documents with shuffled sentences which encourages the model to ignore the external context. In addition, the use of truncated back-propagation (typically at 20 words to enable efficient training) limits the ability of the model to learn long-distance dependencies. We will add discussion in the paper.
>
>         * (More comparisons) We performed an additional experiment using ELMo-pool on document segmentation. For document segmentation, using fixed ELMo-pool representations as block features for a document-level LSTM results in 42.3 F1. This is significantly lower than the 54.9 by L+R-LM and 51.9 by Global-MLM.  We are working on evaluating ELMo-pool on summarization as well.

---

> > ### Author Response · Authors · 2018-11-27
> > **Paper updated**
> >
> > We have updated the paper and addressed several issues pointed out by the reviewers. Specifically, we add more comparisons to Elmo-Pool and skip-thought as suggested.
> >
> > * We added comparisons to ELMo-Pool and skip-thought vectors on all tasks, and our model out-performs these prior methods pretraining sentence embeddings. Note that we reran the segmentation model with ELMo-Pool with different hyper-parameters, and got slightly better results than before. However, it is still under-performs our models.
> >
> > * We improved the baselines (especially the L+R LSTM) and we reran all summarization experiments with models using an additional LSTM layer.
> >
> > * We update and include some more recent results on the summarization tasks.
> >
> > For the future revision, we plan to include more experimental details (which were included in our prior response) in the appendix.

---

### Official Review · AnonReviewer1 · 2018-11-03
**Good incremental work showing the value of pretraining**

**Rating:** 6
**Confidence:** 4

**Review:**

Summary:
This paper proposes to extend the pretraining used for word representations in QA (e.g., ELMO) in the following sense: Instead of just predicting next/previous words in a sentence/paragraph, performing a hierarchical prediction over the whole document, by having a local LSTM and a global LSTM as presented in Fig. 1 + the idea of masked language model. Authors show meaningful improvements in 3 tasks that require document level understanding: extractive summarization, document segmentation, and answer passage retrieval for doc level QA.

Pros:
- Good presentation and clear explanations.
- Meaningful improvements in various tasks requiring document level understanding.

Cons:
- Novelty is mainly incremental

Minor comment:
- Use a bigger picture for Fig. 1
- In page 1, Introduction, paragraph 2, line 10, "due the long-distance ..." ==> "due to the long-distance ..."

**********
I would like to thank authors for their feedback. After reading their feedback I still believe that novelty is incremental and would like to keep my score.

---

> ### Author Response · Authors · 2018-11-14
> **Initial Response**
>
> We thank Reviewer 1 for the valuable comments and will update the paper to address the comments.
>
> The novelty of our paper can be summarized into three points:
>
> * While language model pre-training has been studied before, language model pre-training on document-level context has not been studied extensively.
>
> * We extend the language model pre-training framework to learning representations of thousands of tokens through hierarchical models. Previous work has pre-trained non-hierarchical representations of at most hundreds of tokens through language model pre-training.
>
> * We compare the effectiveness of combining pre-trained uni-directional representations versus pre-training bidirectional representations directly, which has not been done before.

---

> > ### Author Response · Authors · 2018-11-27
> > **Paper updated**
> >
> > We have updated the paper and addressed several issues pointed out by the reviewers.
> >
> > * We added comparisons to ELMo-Pool and skip-thought vectors on all tasks, and our model out-performs these prior methods pretraining sentence embeddings. Note that we reran the segmentation model with ELMo-Pool with different hyper-parameters, and got slightly better results than before. However, it is still under-performs our models.
> >
> > * We improved the baselines (especially the L+R LSTM) and we reran all summarization experiments with models using an additional LSTM layer.
> >
> > * We update and include some more recent results on the summarization tasks.
> >
> > For the future revision, we plan to include more experimental details (which were included in our prior response) in the appendix.

---

### Official Review · AnonReviewer3 · 2018-11-05
**Reasonable method, but not too much novelty**

**Rating:** 6
**Confidence:** 4

**Review:**

Reasonable method, but not too much novelty

[Summary]

The paper proposed techniques to pretrain two-layer hierarchical bi-directional or single-directional LSTM networks for language processing tasks. In particular, the paper uses the word prediction, either for the next work or randomly missing words, as the self-supervised pretraining tasks. The main idea is to not only train text embedding using context from the same sentence but also take the embedding of the surrounding sentences into account, where the sentence embedding is also context-aware. Experiments are done for document segmentation, answer passage retrieval, extractive document summary.

[Pros]

1.	The idea of considering across-sentence/paragraph context for text embedding learning is very reasonable.
2.	The random missing-word completion is also a reasonable self-supervised learning task.
3.	The results are consistently encouraging across all three task. And the performance for “answer passage retrieval” is especially good.

[Cons]

1.	The ideas of predicting the next word (L+R-LM) or missing words (mask-LM) have been around and widely used for a long time. Apply this idea to an two-layer hierarchical LSTM is a straightforward extension of this existing idea.
2.	For document segmentation, no comparison with other methods is provided. For extractive document summary, the performance difference between the proposed method and the previous methods are very minor.
3.	Importantly, the experiments can be stronger if the learned embedding can be successfully applied to more fundamental tasks, such as document classification and retrieval.

Overall, the paper proposed a reasonable method, but the significance of the paper can be better justified by more solid experiments.

---

> ### Author Response · Authors · 2018-11-14
> **Initial Response**
>
> We thank Reviewer 3 for the valuable comments.
>
> [Overall] We would like to clarify the paper and point out that the main novelty of the paper is to ask the research question: “what is the value of pretraining document-level hierarchical models with document-level context?” While language model pre-training has been studied before, language model pretraining on document-level context has not been studied extensively.
>
> [1] While objective functions such as left-to-right LM next word prediction or missing word prediction have been proposed before, no prior work has applied them to pretrain hierarchical document-level models. Unlike prior work like word2vec or Collobert et al.-11 that used missing word prediction to pretrain only context-independent word embeddings, we used this and uni-directional LM objectives to pre-train millions of parameters of a hierarchical document-level representation which contextualizes text segment representations with respect to thousands of tokens in the document.
>
> [2]  (Document segmentation task) We performed an additional experiment using ELMo-pool. Using fixed ELMo-pool representations as block features for a document-level LSTM results in 42.3 F1 on the document segmentation task. This is significantly lower than the 54.9 by L+R-LM and 51.9 by Global-MLM.
>
> [3] Note that, unlike other pre-training methods for document classification tasks, our model does *not* generate a single vector for an input document; this is why we did not apply the model to document classification/retrieval tasks. The focus of the paper is on pretraining hierarchical representations. The state-of-the-art models for segmentation and document summarization use hierarchical models,  because they require representations of individual sentences or paragraphs in document-level context. Our focus is on improving such contextual representations and thus choose these tasks to evaluate the effectiveness of our approach.

---

> > ### Author Response · Authors · 2018-11-27
> > **Paper updated**
> >
> > We have updated the paper and addressed several issues pointed out by the reviewers. Specifically, we add more comparisons for the segmentation experiments as suggested.
> >
> > * We added comparisons to ELMo-Pool and skip-thought vectors on all tasks, and our model out-performs these prior methods pretraining sentence embeddings. Note that we reran the segmentation model with ELMo-Pool with different hyper-parameters, and got slightly better results than before. However, it is still under-performs our models.
> >
> > * We improved the baselines (especially the L+R LSTM) and we reran all summarization experiments with models using an additional LSTM layer.
> >
> > * We update and include some more recent results on the summarization tasks.
> >
> > For the future revision, we plan to include more experimental details (which were included in our prior response) in the appendix.

---

### Meta-Review · Area_Chair1 · 2018-12-13
**Reasonable results, limited novelty**

**Confidence:** 4
**Recommendation:** Reject

**Metareview:**

This paper proposes to pre-train hierarchical document representations for use in downstream tasks. All reviewers agreed that the results were reasonable.

However, the methodological novelty is limited. While I believe there is a place for solid empirical results, even if not incredibly novel, there is also little qualitative or quantitative analysis to shed additional insights.

Given the high quality bar for ICLR, I can't recommend the paper for acceptance at this time.